## PROCEEDINGS A

mathematical modelling, differential equations, computational mathematics

social dilemma, vaccination game, sir model

**Author for correspondence:**
Kazuki Kuga
e-mail: kuga@kyudai.jp

# Pair approximation model for the vaccination game: predicting the dynamic process of epidemic spread and individual actions against contagion

Kazuki Kuga[1], Masaki Tanaka[1] and Jun Tanimoto[1,2]

[1]Interdisciplinary Graduate School of Engineering Sciences, and [2]Faculty of Engineering Sciences, Kyushu University, Kasuga-koen, Kasuga-shi, Fukuoka 816-8580, Japan

KK, 0000-0002-6553-6936; MT, 0000-0003-0943-6890

We successfully establish a theoretical framework of pairwise approximation for the vaccination game in which both the dynamic process of epidemic spread and individual actions in helping prevent social behaviours are quantitatively evaluated. In contrast with mean-field approximation, our model captures higher-order effects from neighbours by using an underlying network that shows how the disease spreads and how individual decisions evolve over time. This model considers not only imperfect vaccination but also intermediate protective measures other than vaccines. Our analytical predictions are validated by multi-agent simulation results that estimate random regular graphs at varying degrees.

## 1. Introduction

In today's globalized world, all kinds of infectious diseases, such as measles, smallpox, pertussis, flu and Ebola, can spread rapidly around the world and negatively affect the lives of people. Vaccination is one of the most effective measures for preventing the transmission of infectious diseases and for reducing morbidity and mortality rates [1,2]. However, voluntary vaccination policies create a social dilemma [3,4], namely, the 'vaccination dilemma' or 'paradox of epidemiology'.

As a result, individuals often hesitate to get themselves vaccinated. When herd immunity is achieved in a community with increased vaccination coverage (VC), non-vaccinators can avoid infections and vaccination costs. However, herd immunity is inevitably destroyed because non-vaccinators completely depend on the efforts of vaccinators. Therefore, the actual VC will not be able to satisfy demand.

To model this vaccination dilemma, researchers have studied vaccination games, which can predict the dynamics of (i) epidemic spread in a complex social network and (ii) decision making on whether to undergo vaccination depending on the status of the epidemic. Epidemic dynamics are predicted using mathematical epidemiological models, such as the susceptible–infected–recovered (SIR) model, whereas decision making is modelled on evolutionary game theory. Bauch et al. [5] and Fu et al. [6] published pioneering works on the vaccination game. To quantitatively investigate the multiple effects on vaccination behaviour, a significant number of researchers have studied various frameworks [7–13]. Most studies on the vaccination game have relied on multi-agent simulation (MAS), which allows for a more flexible and realistic modelling approach. In addition to the MAS approach, Fu et al. [6] proposed a mathematical framework for a mixed-population vaccination game that assumes perfect vaccination. A theoretical approach based on a set of ordinary differential equations (ODEs) can be a powerful tool for explicitly demonstrating the dynamics of both epidemic spread and human decision making. In most recent study, Steinegger et al. [14] provided a theoretical framework that allows to incorporate the structure of the network in this kind of disease-driven dilemmas using Markovian equations.

Most studies on the vaccination game have assumed that vaccinations provide perfect immunity to each vaccinator. In reality, vaccinations can only impart partial protection against many infectious diseases, such as measles, influenza, malaria and HIV. In addition to vaccinations, there are other protective measures from infectious diseases, such as mask-wearing, gargling and hand washing, which are called intermediate protective measures. Although these protective measures come at a more reasonable cost than vaccinations, they cannot block the transmission of infections to the body as effectively as vaccinations. Therefore, the stochastic effects of imperfect vaccination and intermediate protective measures need to be considered. On the basis of this background, Cardillo et al. [15] analysed the effects of imperfect vaccination on immunization behaviour in Erdős–Rényi random graph (ER–RG) and Barabasi–Albert scale-free (BA–SF) networks by using the MAS approach. Iwamura et al. [16] and Ida and Tanimoto [17] used MAS to investigate the effect of intermediate protective measures on square lattice and BA–SF networks. Wu et al. [18] and Kuga and Tanimoto [19] developed a new mathematical framework of the vaccination game that considered imperfect vaccination in an infinite and well-mixed population corresponding with a perfect graph by using mean-field approximation. Furthermore, they proposed a mathematical model for reproducing the vaccination game in complex networks corresponding to ER–RG and BA–SF networks by using degree distribution [20]. Several studies have recently used the same concept and investigated the multiple effects of imperfect vaccination and other parameters on vaccination behaviour [21–32]. Therefore, the modelling and analysis of the vaccination game can be enriched by the abundant quantity and quality of studies that have followed a MAS or theoretical approach.

While highlighting the theoretical approach, it should be noted that most mathematical studies on the vaccination game have focused on the so-called mean-field approximation, which is a single-order approximation for predicting the dynamics of epidemics and decision making, despite being extended to some heterogeneous topologies, such as infinite-sized graphs with degree distributions obeying the power law or Poisson distribution [20,30]. However, mean-field approximation has substantial limitations because it ignores higher order effects and assumes that one's neighbour can always be defined as a socially averaged hypothetical neighbour, which can never be true in a real-world, complex social network. This defect means that predictions by mean-field approximation are slightly different from those by MAS, which can capture predictions to a reasonable extent. As an alternative to mean-field approximation, the current study uses pair approximation, which has been widely applied in other fields. For example, in the spatial version of the Prisoner's Dilemma, which is an archetype model for quantifying network

reciprocity, Fu *et al*. [33,34] proposed a pair approximation model as a coevolutionary Prisoner's Dilemma game. Given that the pair approximation model is more effective and accurate than the mean-field approximation model, it can also be applied to mathematical epidemiological models. In fact, the pair approximation model can provide explicit treatment of the epidemic process at both the node and link levels. Keeling [35] built a pair approximation of a SIR epidemic model, which estimated topological clusters and discussed the basic reproduction number and the final epidemic size. Bauch [36] established a pair approximation of a susceptible–infected– susceptible epidemic model and analysed its basic reproduction number. Tomé and De Oliveira [37] also developed a pair approximation of SIR and susceptible–exposed–infected epidemic models on a Cayley tree and solved the time-evolutionary equations to determine the final and time-dependent properties. However, no study has highlighted how pair approximation can be applied to the vaccination game, where the effect of protective measures, including vaccination, is modelled as a stochastic process. Additionally, the pair approximation model for vaccination game would benefit the studies focusing on the effects of subsided-vaccination policy on epidemic spreading [31,38,39].

This study seeks to develop a mathematical model for the vaccination game by using SIR/V models that recreate either imperfect vaccinations or intermediate protective measures that rely on pair approximation. To demonstrate examples of numerical simulation, we focus on the dynamics of epidemic spread and decision making in regular random networks of varying degrees.

The rest of the paper is organized as follows. Section 2 of this paper presents a description of the model and assumptions for deductive analysis. Section 3 provides the deductive results and the discussions, which are fairly validated by a series of MAS results. Section 4 summarizes our findings.

## 2. Model set-up and results

## (a) Epidemic models

The pair approximation SIR model [35,37] was used to accurately model epidemic spread. We integrated the pair approximation SIR model with the concept of effectiveness and efficiency models, which can reproduce imperfect vaccination and intermediate protective measures [19,20]. The electronic supplementary material describes the fundamental formulation when the pair approximation is applied to SIR dynamics.

### (i) Effectiveness of vaccination

In the effectiveness model, a vaccinated population is separated into two classes: individuals with perfect immunity and non-immune individuals. The population is subdivided into the following: non-vaccinated susceptible individual $S_N$, non-vaccinated infected individual $I_N$, non-vaccinated recovered individual $R_N$, vaccinated susceptible individual $S_V$, vaccinated infected individual $I_V$, vaccinated recovered individual $R_V$ and vaccinated individual with perfect immunity $P_V$. Let the effectiveness of the vaccination and the vaccination coverage be $e$ ($0 \leq e \leq 1$) and $x$, respectively. The fraction of vaccinated individuals with perfect immunity $[P_V](t)$ must be $ex$, whereas that of non-immune individuals $[S_V](t) + [I_V](t) + [R_V](t)$ is $(1-e)x$. On the basis of the above assumptions, the dynamics of the SIR/V model with imperfect vaccination can be described by the following ODEs:

$$\frac{\mathrm{d}}{\mathrm{d}t}[S_N](t) = -\beta([S_N I_N](t) + [S_N I_V](t)), \tag{2.1}$$

$$\frac{\mathrm{d}}{\mathrm{d}t}[S_V](t) = -\beta([S_V I_N](t) + [S_V I_V](t)), \tag{2.2}$$

$$\frac{d}{dt}[P_V](t) = 0, \tag{2.3}$$

$$\frac{d}{dt}[S_N S_N](t) = -2\beta[S_N S_N](t)(Q(I_N|S_N S_N) + Q(I_V|S_N S_N)), \tag{2.4}$$

$$\frac{d}{dt}[S_N R_N](t) = \gamma[S_N I_N](t) - \beta[S_N R_N](t)(Q(I_N|S_N R_N) + Q(I_V|S_N R_N)), \tag{2.5}$$

$$\frac{d}{dt}[S_V S_V](t) = -2\beta[S_V S_V](t)(Q(I_N|S_V S_V) + Q(I_V|S_V S_V)), \tag{2.6}$$

$$\frac{d}{dt}[S_V R_V](t) = \gamma[S_V I_V](t) - \beta[S_V R_V](t)(Q(I_N|S_V R_V) + Q(I_V|S_V R_V)), \tag{2.7}$$

$$\frac{d}{dt}[S_V P_V](t) = -\beta[S_V P_V](t)(Q(I_N|S_V P_V) + Q(I_V|S_V P_V)), \tag{2.8}$$

$$\frac{d}{dt}[P_V P_V](t) = 0, \tag{2.9}$$

$$\frac{d}{dt}[S_N S_V](t) = -\beta[S_N S_V](t)(Q(I_N|S_N S_V) + Q(I_V|S_N S_V))$$
$$-\beta[S_N S_V](t)(Q(I_N|S_V S_N) + Q(I_V|S_V S_N)), \tag{2.10}$$

$$\frac{d}{dt}[S_N R_V](t) = \gamma[S_N I_V](t) - \beta[S_N R_V](t)(Q(I_N|S_N R_V) + Q(I_V|S_N R_V)), \tag{2.11}$$

$$\frac{d}{dt}[S_V R_N](t) = \gamma[S_V I_N](t) - \beta[S_V R_N](t)(Q(I_N|S_V R_N) + Q(I_V|S_V R_N)) \tag{2.12}$$

and $$\frac{d}{dt}[S_N P_V](t) = -\beta[S_N P_V](t)(Q(I_N|S_N P_V) + Q(I_V|S_N P_V)). \tag{2.13}$$

The above set of dynamic equations should be assumed to have the following set of initial conditions: $[S_N](0) = 1 - x$, $[S_V](0) = (1-e)x$, $[P_V](0) = ex$, $[I_N](0) \sim 0$, $[I_V](0) \sim 0$, $[R_N](0) = 0$, $[R_V](0) = 0$, $[S_N S_N](0) = Q(1 - x - \alpha)$, $[S_N S_V](0) = Q(1-e)\alpha$, $[S_N P_V](0) = Qe\alpha$, $[S_V S_V](0) = Q(1-e)^2(x - \alpha)$, $[S_V P_V](0) = Qe(1-e)(x - \alpha)$, $[P_V P_V](0) = Qe^2(x - \alpha)$.

Here, $\alpha$ is the vaccinator–non-vaccinator connection coefficient, which can be said dissortativity observed at the initial moment of every season (time-evolved in repeating seasons), and is less than $x$ and $1 - x$. If the homogeneous distribution of vaccinator and non-vaccinated was assumed, $\alpha = x(1 - x)$.

The following constraints are required:

$$[S_N](t) + [I_N](t) + [R_N](t) + [S_V](t) + [I_V](t) + [R_V](t) + [P_V](t) = 1, \tag{2.14}$$

$$[S_N S_N](t) + [S_N I_N](t) + [S_N R_N](t) + [S_N S_V](t) + [S_N I_V](t) + [S_N R_V](t)$$
$$+ [S_N P_V](t) = Q[S_N](t) \tag{2.15}$$

and $$[S_N S_V](t) + [S_V I_N](t) + [S_V R_N](t) + [S_V S_V](t) + [S_V I_V](t)$$
$$+ [S_V R_V](t) + [S_V P_V](t) = Q[S_V](t). \tag{2.16}$$

Solving the above set of equations, we obtain the following equations (the detailed mathematical manipulations are shown in the electronic supplementary material):

$$[S_N R_N](t) + [S_N R_V](t) = Qr(1-x)\left(\left(\frac{[S_N](t)}{1-x}\right)^\mu - \frac{[S_N](t)}{1-x}\right), \tag{2.17}$$

$$[S_V R_N](t) + [S_V R_V](t) = Qr(1-e)x\left(\left(\frac{[S_V](t)}{(1-e)x}\right)^\mu - \frac{[S_V](t)}{(1-e)x}\right), \tag{2.18}$$

$$[S_N S_N](t) = Q(1 - x - \alpha)\left(\frac{[S_N](t)}{1-x}\right)^{2\mu}, \tag{2.19}$$

$$[S_V S_V](t) = Q(1-e)^2(x-\alpha)\left(\frac{[S_V](t)}{(1-e)x}\right)^{2\mu}, \tag{2.20}$$

$$[S_N S_V](t) = Q(1-e)\alpha\left(\frac{[S_N](t)}{1-x}\right)^{\mu}\left(\frac{[S_V](t)}{(1-e)x}\right)^{\mu}, \tag{2.21}$$

$$[S_N P_V](t) = Qe\alpha\left(\frac{[S_N](t)}{1-x}\right)^{\mu} \tag{2.22}$$

and
$$[S_V P_V](t) = Qe(1-e)(x-\alpha)\left(\frac{[S_V](t)}{(1-e)x}\right)^{\mu}. \tag{2.23}$$

At the steady state ($t \to \infty$), the constraints in equation (2.14) to (2.16) can be rewritten as follows:

$$[S_N](\infty) + [R_N](\infty) + [S_V](\infty) + [R_V](\infty) + [P_V](\infty) = 1, \tag{2.24}$$

$$[S_N S_N](\infty) + [S_N R_N](\infty) + [S_N S_V](\infty) + [S_N R_V](\infty) + [S_N P_V](\infty) = Q[S_N](\infty) \tag{2.25}$$

and $\quad [S_N S_V](\infty) + [S_V R_N](\infty) + [S_V S_V](\infty) + [S_V R_V](\infty) + [S_V P_V](\infty) = Q[S_V](\infty). \tag{2.26}$

Substituting $[S_N R_N](\infty) + [S_N R_V](\infty)$, $[S_N S_N](\infty)$, $[S_N S_V](\infty)$, and $[S_N P_V](\infty)$ from equations (2.17), (2.19), (2.21) and (2.22) into equation (2.25) yields as follows:

$$(1-x-\alpha)\left(\frac{[S_N](\infty)}{1-x}\right)^{2\mu} + (1-e)\alpha\left(\frac{[S_N](\infty)}{1-x}\right)^{\mu}\left(\frac{[S_V](\infty)}{(1-e)x}\right)^{\mu}$$
$$+ r(1-x)\left(\left(\frac{[S_N](\infty)}{1-x}\right)^{\mu} - \frac{[S_N](\infty)}{1-x}\right) + e\alpha\left(\frac{[S_N](\infty)}{1-x}\right)^{\mu} = (1-x)\frac{[S_N](\infty)}{1-x}. \tag{2.27}$$

Substituting $[S_V R_N](\infty) + [S_V R_V](\infty)$, $[S_V S_V](\infty)$, $[S_N S_V](\infty)$, and $[S_V P_V](\infty)$ from equations (2.18), (2.20), (2.21) and (2.23) into equation (2.26) yields as follows:

$$\alpha\left(\frac{[S_N](t)}{1-x}\right)^{\mu}\left(\frac{[S_V](t)}{(1-e)x}\right)^{\mu} + rx\left(\left(\frac{[S_V](t)}{(1-e)x}\right)^{\mu} - \frac{[S_V](t)}{(1-e)x}\right)$$
$$+ (1-e)(x-\alpha)\left(\frac{[S_V](t)}{(1-e)x}\right)^{2\mu} + e(x-\alpha)\left(\frac{[S_V](t)}{(1-e)x}\right)^{\mu} = x\frac{[S_V](t)}{(1-e)x}. \tag{2.28}$$

By defining $p = ([S_N](\infty)/1-x)^{1/Q}$, $q = ([S_V](\infty)/(1-e)x)^{1/Q}$ and taking into account the definition of $\mu = (Q-1)/Q$, we can write equations (2.27) and (2.28) as follows:

$$(1-x-\alpha)p^{Q-1} - (1+r)(1-x)p + r(1-x) + e\alpha + (1-e)\alpha q^{Q-1} = 0 \tag{2.29}$$

and
$$(1-e)(x-\alpha)q^{Q-1} - (1+r)xq + rx + e(x-\alpha) + \alpha p^{Q-1} = 0. \tag{2.30}$$

The final fractions are expressed as follows:

$$[S_N](\infty) = (1-x)p^Q, \tag{2.31}$$

$$[S_V](\infty) = (1-e)xq^Q, \tag{2.32}$$

$$[P_V](\infty) = ex, \tag{2.33}$$

$$[R_N](\infty) = 1 - x - [S_N](\infty) = (1-x)(1-p^Q) \tag{2.34}$$

and
$$[R_V](\infty) = (1-e)x - [S_V](\infty) = (1-e)x(1-q^Q). \tag{2.35}$$

The other final pairs are hypothetically calculated as follows:

$$[S_N R_N](\infty) = Qr(1-x)(p^{Q-1} - p^Q)\frac{[R_N](\infty)}{[R_N](\infty) + [R_V](\infty)}, \tag{2.36}$$

$$[S_N R_N](\infty) = Qr(1-x)(p^{Q-1} - p^Q)\frac{[R_V](\infty)}{[R_N](\infty) + [R_V](\infty)}, \tag{2.37}$$

$$[S_V R_N](\infty) = Qr(1-e)x(q^{Q-1} - q^Q)\frac{[R_N](\infty)}{[R_N](\infty) + [R_V](\infty)}, \tag{2.38}$$

$$[S_V R_N](\infty) = Qr(1-e)x(q^{Q-1} - q^Q)\frac{[R_V](\infty)}{[R_N](\infty) + [R_V](\infty)}, \tag{2.39}$$

$$[R_N R_N](\infty) = [S_N S_N](0) - [S_N S_N](\infty) - 2[S_N R_N](\infty), \tag{2.40}$$

$$[R_V R_V](\infty) = [S_V S_V](0) - [S_V S_V](\infty) - 2[S_V R_V](\infty), \tag{2.41}$$

$$[R_N R_V](\infty) = [S_N S_V](0) - [S_N S_V](\infty) - [S_N R_V](\infty) - [S_V R_N](\infty), \tag{2.42}$$

$$[R_N P_V](\infty) = [S_N P_V](0) - [S_N P_V](\infty) = Qex(1-x)(1-p^{Q-1}) \tag{2.43}$$

and
$$[R_V P_V](\infty) = [S_V P_V](0) - [S_V P_V](\infty) = Qe(1-e)x^2(1-q^{Q-1}). \tag{2.44}$$

## (ii) Efficiency of intermediate protective measures

Let the efficiency of an intermediate protective measure to avoid infection be $\eta$ $(0 \le \eta \le 1)$, i.e. the extent in which the protective measure decreases the probability of infection. In the following efficiency model formulation, we temporarily regard the vaccinated state $V$ as the state prepared with an intermediate protective measure for comparison with the effectiveness model. A non-vaccinated susceptible individual $S_N$ (more precisely, an individual not prepared with intermediate protective measures) may become infected if he/she is exposed to infectious individuals with a disease transmission rate of $\beta$. A vaccinated (i.e. prepared) individual $S_V$ who is taking intermediate protective measures may also become infectious with $(1-\eta)\beta$. On the basis of the above assumptions and by taking into account the intermediate protective measures, we describe the following differential equations for the dynamics of epidemic spread:

$$\frac{d}{dt}[S_N](t) = -\beta([S_N I_N](t) + [S_N I_V](t)), \tag{2.45}$$

$$\frac{d}{dt}[S_V](t) = -(1-\eta)\beta([S_V I_N](t) + [S_V I_V](t)), \tag{2.46}$$

$$\frac{d}{dt}[S_N S_N](t) = -2\beta[S_N S_N](t)(Q(I_N|S_N S_N) + Q(I_V|S_N S_N)), \tag{2.47}$$

$$\frac{d}{dt}[S_N R_N](t) = \gamma[S_N I_N](t) - \beta[S_N R_N](t)(Q(I_N|S_N R_N) + Q(I_V|S_N R_N)), \tag{2.48}$$

$$\frac{d}{dt}[S_V S_V](t) = -2(1-\eta)\beta[S_V S_V](t)(Q(I_N|S_V S_V) + Q(I_V|S_V S_V)), \tag{2.49}$$

$$\frac{d}{dt}[S_V R_V](t) = \gamma[S_V I_V](t) - (1-\eta)\beta[S_V R_V](t)(Q(I_N|S_V R_V) + Q(I_V|S_V R_V)), \tag{2.50}$$

$$\frac{d}{dt}[S_N S_V](t) = -\beta[S_N S_V](t)(Q(I_N|S_N S_V) + Q(I_V|S_N S_V))$$
$$- (1-\eta)\beta[S_N S_V](t)(Q(I_N|S_V S_N) + Q(I_V|S_V S_N)), \tag{2.51}$$

$$\frac{d}{dt}[S_N R_V](t) = \gamma[S_N I_V](t) - \beta[S_N R_V](t)(Q(I_N|S_N R_V) + Q(I_V|S_N R_V)) \tag{2.52}$$

and
$$\frac{d}{dt}[S_V R_N](t) = \gamma[S_V I_N](t) - (1-\eta)\beta[S_V R_N](t)(Q(I_N|S_V R_N) + Q(I_V|S_V R_N)). \tag{2.53}$$

The above set of equations are assumed to have the following initial conditions: $[S_N](0) = 1 - x$, $[S_V](0) = x$, $[I_N](0) \sim 0$, $[I_V](0) \sim 0$, $[R_N](0) = 0$, $[R_V](0) = 0$, $[S_N S_N](0) = Q(1-x-\alpha)$, $[S_N S_V](0) = Q\alpha$, and $[S_V S_V](0) = Q(x-\alpha)$.

The following constraints are required:

$$[S_N](t) + [I_N](t) + [R_N](t) + [S_V](t) + [I_V](t) + [R_V](t) = 1, \tag{2.54}$$

$$[S_N S_N](t) + [S_N I_N](t) + [S_N R_N](t) + [S_N S_V](t) + [S_N I_V](t) + [S_N R_V](t) = Q[S_N](t) \tag{2.55}$$

and
$$[S_N S_V](t) + [S_V I_N](t) + [S_V R_N](t) + [S_V S_V](t) + [S_V I_V](t) + [S_V R_V](t) = Q[S_V](t). \tag{2.56}$$

Solving the above set of equations, we obtain the following equations (the detailed mathematical manipulations are shown in the electronic supplementary material):

$$[S_N R_N](t) + [S_N R_V](t) = Qr(1-x)\left(\left(\frac{[S_N](t)}{1-x}\right)^\mu - \frac{[S_N](t)}{1-x}\right), \tag{2.57}$$

$$[S_V R_N](t) + [S_V R_V](t) = \frac{Qrx}{1-\eta}\left(\left(\frac{[S_V](t)}{x}\right)^\mu - \frac{[S_V](t)}{x}\right), \tag{2.58}$$

$$[S_N S_N](t) = Q(1-x-\alpha)\left(\frac{[S_N](t)}{1-x}\right)^{2\mu}, \tag{2.59}$$

$$[S_V S_V](t) = Q(x-\alpha)\left(\frac{[S_V](t)}{x}\right)^{2\mu} \tag{2.60}$$

and
$$[S_N S_V](t) = Q\alpha\left(\frac{[S_N](t)}{1-x}\right)^\mu\left(\frac{[S_V](t)}{x}\right)^\mu. \tag{2.61}$$

At the steady state ($t \to \infty$), the constraints in equations (2.54) to (2.56) can be rewritten as follows:

$$[S_N](\infty) + [R_N](\infty) + [S_V](\infty) + [R_V](\infty) = 1, \tag{2.62}$$

$$[S_N S_N](\infty) + [S_N R_N](\infty) + [S_N S_V](\infty) + [S_N R_V](\infty) = Q[S_N](\infty) \tag{2.63}$$

and
$$[S_V S_N](\infty) + [S_V R_N](\infty) + [S_V S_V](\infty) + [S_V R_V](\infty) = Q[S_V](\infty). \tag{2.64}$$

Substituting $[S_N R_N](\infty) + [S_N R_V](\infty)$, $[S_N S_N](\infty)$ and $[S_N S_V](\infty)$ from equations (2.57), (2.59) and (2.61) into equation (2.63) yields the following:

$$Q(1-x-\alpha)\left(\frac{[S_N](\infty)}{1-x}\right)^{2\mu} + Qr(1-x)\left(\left(\frac{[S_N](\infty)}{1-x}\right)^\mu - \frac{[S_N](\infty)}{1-x}\right)$$
$$+ Q\alpha\left(\frac{[S_N](\infty)}{1-x}\right)^\mu\left(\frac{[S_V](\infty)}{x}\right)^\mu = Q(1-x)\frac{[S_N](\infty)}{1-x}. \tag{2.65}$$

Substituting, $[S_N R_N](\infty) + [S_N R_V](\infty)$, $[S_V S_V](\infty)$ and $[S_N S_V](\infty)$ from equations (2.58), (2.60) and (2.61) into equation (2.64) yields the following:

$$Q(x-\alpha)\left(\frac{[S_V](\infty)}{x}\right)^{2\mu} + \frac{Qrx}{1-\eta}\left(\left(\frac{[S_V](\infty)}{x}\right)^\mu - \frac{[S_V](\infty)}{x}\right)$$
$$+ Q\alpha\left(\frac{[S_N](\infty)}{1-x}\right)^\mu\left(\frac{[S_V](\infty)}{x}\right)^\mu = Qx\frac{[S_V](\infty)}{x}. \tag{2.66}$$

By defining $p = ([S_N](\infty)/1-x)^{1/Q}$ and $q = ([S_V](\infty)/x)^{1/Q}$ and by taking into account the definition of $\mu$, we can write equations (2.65) and (2.66) as follows:

$$(1-x-\alpha)p^{Q-1} - (1+r)(1-x)p + r(1-x) + \alpha q^{Q-1} = 0 \tag{2.67}$$

and

$$(x-\alpha)q^{Q-1} - \left(1+\frac{r}{1-\eta}\right)xq + \frac{rx}{1-\eta} + \alpha p^{Q-1} = 0. \tag{2.68}$$

The final fractions are expressed as follows:

$$[S_N](\infty) = (1-x)p^Q, \tag{2.69}$$

$$[S_V](\infty) = xq^Q, \tag{2.70}$$

$$[R_N](\infty) = 1-x-[S_N](\infty) = (1-x)(1-p^Q) \tag{2.71}$$

and
$$[R_V](\infty) = x-[S_V](\infty) = x(1-q^Q). \tag{2.72}$$

**Table 1.** Payoff structure determined at the end of an epidemic season.

| strategy | V | | NV | |
|---|---|---|---|---|
| state | HV | IV | SFR | FFR |
| payoff | $-C_r$ | $-C_r - 1$ | 0 | $-1$ |

The other final pairs are hypothetically calculated as follows:

$$[S_N R_N](\infty) = Qr(1-x)\left(p^{Q-1} - p^Q\right) \frac{[R_N](\infty)}{[R_N](\infty) + [R_V](\infty)}, \qquad (2.73)$$

$$[S_N R_V](\infty) = Qr(1-x)\left(p^{Q-1} - p^Q\right) \frac{[R_V](\infty)}{[R_N](\infty) + [R_V](\infty)}, \qquad (2.74)$$

$$[S_V R_N](\infty) = \frac{Qrx}{1-\eta}\left(q^{Q-1} - q^Q\right) \frac{[R_N](\infty)}{[R_N](\infty) + [R_V](\infty)}, \qquad (2.75)$$

$$[S_V R_V](\infty) = \frac{Qrx}{1-\eta}\left(q^{Q-1} - q^Q\right) \frac{[R_V](\infty)}{[R_N](\infty) + [R_V](\infty)}, \qquad (2.76)$$

$$[R_N R_N](\infty) = [S_N S_N](0) - [S_N S_N](\infty) - 2[S_N R_N](\infty) \qquad (2.77)$$

$$[R_V R_V](\infty) = [S_V S_V](0) - [S_V S_V](\infty) - 2[S_V R_V](\infty) \qquad (2.78)$$

and

$$[R_N R_V](\infty) = [S_N S_V](0) - [S_N S_V](\infty) - [S_N R_V](\infty) - [S_V R_N](\infty). \qquad (2.79)$$

## (b) Payoff structure

An epidemic season continues until the number of infected individuals is reduced to zero. If individuals are infected, they pay the cost of infection $C_i$. When individuals are vaccinated or take intermediate protective measures, they pay the cost $C_v$. To clearly evaluate each individual's payoff, we define the relative cost of vaccination, namely, $C_r = C_v / C_i$ ($0 \le C_r \le 1$; $C_i = 1$). Consequently, depending on the cost burden, the state of the individual is classified as follows: (i) a healthy vaccinator (HV), who pays $-C_r$; (ii) an infected vaccinator (IV), who pays $-C_r - 1$; (iii) a successful free rider (SFR), who pays nothing and (iv) a failed free rider (FFR), who pays $-1$. There are two strategies that can be chosen: taking the vaccination (or intermediate protective measures) (V) and not taking the vaccination (NV). Table 1 summarizes the payoff on the basis of whether the approach was V (either vaccination or intermediate protective measure) or NV and whether the individual is healthy or infected.

On the basis of the expected payoff, the average social payoff $\langle \pi \rangle$ is formulated as follows: (Effectiveness model)

$$\langle \pi \rangle = -C_r([S_V](\infty) + [P_V](\infty)) - (C_r + 1)[R_V](\infty) - [R_N](\infty)$$

$$= -C_r((1-e)xp^Q + ex) - (C_r + 1)(1-e)x(1 - p^Q) - (1-x)(1 - p^Q). \qquad (2.80)$$

(Efficiency model)

$$\langle \pi \rangle = -C_r[S_V](\infty) - (C_r + 1)[R_V](\infty) - [R_N](\infty)$$

$$= -C_r x q^Q - (C_r + 1)x(1 - q^Q) - (1-x)(1 - p^Q). \qquad (2.81)$$

## (c) Strategy updating

Within the context of the vaccination game, an individual can change his/her strategy at the end of every epidemic season by deciding whether to rely on a provision and by reflecting on the

payoff in the previous epidemic season. In this study, we consider the strategy-updating approach proposed by Fu *et al.* [6], which is called 'individual-based risk assessment' (IB-RA).

In IB-RA, for two-strategy and two-player ($2 \times 2$) games, an individual can update their strategy by randomly selecting neighbouring individuals stochastically and copying them. The individual decides whether to adopt that neighbour's strategy by using pairwise Fermi updating [6], i.e. individual $i$ adopts the selected neighbour $j$'s strategy with a probability:

$$P(s_i \leftarrow s_j) = \frac{1}{1 + \exp[-(\pi_j - \pi_i)/\kappa]}, \tag{2.82}$$

where $s_i$ is the strategy of $i$, $\pi_i$ is $i$'s payoff in the previous season and parameter $\kappa > 0$ characterizes the strength of selection (the sensitivity of individuals to differences in their payoffs); a smaller $\kappa$ indicates that an individual is more sensitive to a payoff difference. We set $\kappa = 0.1$ on the basis of [6].

The transition probability affecting the dynamics of $x$, which should be considered in light of the IB-RA rule, is covered by one of the following eight cases:

$$
\begin{cases}
P(HV \leftarrow SFR) = \dfrac{1}{1 + \exp[-(0 - (-C_r))/\kappa]}, \\[2mm]
P(HV \leftarrow FFR) = \dfrac{1}{1 + \exp[-(-1 - (-C_r))/\kappa]}, \\[2mm]
P(IV \leftarrow SFR) = \dfrac{1}{1 + \exp[-(0 - (-C_r - 1))/\kappa]}, \\[2mm]
P(IV \leftarrow FFR) = \dfrac{1}{1 + \exp[-(-1 - (-C_r - 1))/\kappa]}, \\[2mm]
P(SFR \leftarrow HV) = \dfrac{1}{1 + \exp[-(-C_r - 0)/\kappa]}, \\[2mm]
P(SFR \leftarrow IV) = \dfrac{1}{1 + \exp[-(-C_r - 1 - 0)/\kappa]}, \\[2mm]
P(FFR \leftarrow HV) = \dfrac{1}{1 + \exp[-(-C_r - (-1))/\kappa]}, \\[2mm]
P(FFR \leftarrow IV) = \dfrac{1}{1 + \exp[-(-C_r - 1 - (-1))/\kappa]}.
\end{cases}
\tag{2.83}
$$

After the end of every epidemic season, the vaccination coverage $x$ will increase or decrease. To quantify this evolutionary process, we obtain the following equation for the dynamic system:

Effectiveness model

$$
\begin{aligned}
\frac{dx}{dt} = {} & \frac{[S_N S_V](\infty) + [S_N P_V](\infty)}{Q}(P(SFR \leftarrow HV) - P(HV \leftarrow SFR)) \\[2mm]
& + \frac{[S_N R_V](\infty)}{Q}(P(SFR \leftarrow IV) - P(IV \leftarrow SFR)) \\[2mm]
& + \frac{[S_V R_N](\infty) + [P_V R_N](\infty)}{Q}(P(FFR \leftarrow HV) - P(HV \leftarrow FFR)) \\[2mm]
& + \frac{[R_N R_V](\infty)}{Q}(P(FFR \leftarrow IV) - P(IV \leftarrow FFR)).
\end{aligned}
\tag{2.84}
$$

Efficiency model

$$
\begin{aligned}
\frac{\mathrm{d}x}{\mathrm{d}t} ={}& \frac{[S_N S_V](\infty)}{Q}(P(SFR \leftarrow HV) - P(HV \leftarrow SFR)) \\
&+ \frac{[S_N R_V](\infty)}{Q}(P(SFR \leftarrow IV) - P(IV \leftarrow SFR)) \\
&+ \frac{[S_V R_N](\infty)}{Q}(P(FFR \leftarrow HV) - P(HV \leftarrow FFR)) \\
&+ \frac{[R_N R_V](\infty)}{Q}(P(FFR \leftarrow IV) - P(IV \leftarrow FFR)).
\end{aligned}
\tag{2.85}
$$

As a result of changing the strategy, the vaccinator–non-vaccinator connection coefficient $\alpha$ will also change. To quantify this evolutionary process, we obtain the following equation for the dynamic system:

Effectiveness model

$$
\begin{aligned}
Q\frac{\mathrm{d}\alpha}{\mathrm{d}t} ={}& [S_N S_N](\infty)P(S_N \rightarrow V|S_N S_N)(1 - P(S_N \rightarrow V|S_N S_N)) \\
&+ [S_N R_N](\infty)\{P(S_N \rightarrow V|S_N R_N)(1 - P(R_N \rightarrow V|S_N R_N)) \\
&+ P(R_N \rightarrow V|S_N R_N)(1 - P(S_N \rightarrow V|S_N R_N))\} \\
&+ [R_N R_N](\infty)P(R_N \rightarrow V|R_N R_N)(1 - P(R_N \rightarrow V|R_N R_N)) \\
&+ [S_V S_V](\infty)P(S_V \rightarrow NV|S_V S_V)(1 - P(S_V \rightarrow NV|S_V S_V)) \\
&+ [S_V R_V](\infty)\{P(S_V \rightarrow NV|S_V R_V)(1 - P(R_V \rightarrow NV|S_V R_V)) \\
&+ P(R_V \rightarrow NV|S_V R_V)(1 - P(S_V \rightarrow NV|S_V R_V))\} \\
&+ [R_V R_V](\infty)P(R_V \rightarrow NV|R_V R_V)(1 - P(R_V \rightarrow NV|R_V R_V)) \\
&+ [S_V P_V](\infty)\{P(S_V \rightarrow NV|S_V P_V)(1 - P(P_V \rightarrow NV|S_V P_V)) \\
&+ P(P_V \rightarrow NV|S_V P_V)(1 - P(S_V \rightarrow NV|S_V P_V))\} \\
&+ [R_V P_V](\infty)\{P(R_V \rightarrow NV|S_V P_V)(1 - P(P_V \rightarrow NV|R_V P_V)) \\
&+ P(P_V \rightarrow NV|R_V P_V)(1 - P(S_V \rightarrow NV|R_V P_V))\} \\
&- [S_N S_V](\infty)\{P(S_N \rightarrow V|S_N S_V)(1 - P(S_V \rightarrow NV|S_N S_V)) \\
&+ P(S_V \rightarrow NV|S_N S_V)(1 - P(S_N \rightarrow V|S_N S_V))\} \\
&- [S_N R_V](\infty)\{P(S_N \rightarrow V|S_N R_V)(1 - P(R_V \rightarrow NV|S_N R_V)) \\
&+ P(R_V \rightarrow NV|S_N R_V)(1 - P(S_N \rightarrow V|S_N R_V))\} \\
&- [S_V R_N](\infty)\{P(R_N \rightarrow V|S_V R_N)(1 - P(S_V \rightarrow NV|S_V R_N)) \\
&+ P(S_V \rightarrow NV|S_V R_N)(1 - P(R_N \rightarrow V|S_V R_N))\} \\
&- [R_N R_V](\infty)\{P(R_N \rightarrow V|R_N R_V)(1 - P(R_V \rightarrow NV|R_N R_V)) \\
&+ P(R_V \rightarrow NV|R_N R_V)(1 - P(R_N \rightarrow V|R_N S_V))\} \\
&- [S_N P_V](\infty)\{P(S_N \rightarrow V|S_N P_V)(1 - P(P_V \rightarrow NV|S_N P_V)) \\
&+ P(P_V \rightarrow NV|S_N P_V)(1 - P(S_N \rightarrow V|S_N P_V))\} \\
&- [R_N P_V](\infty)\{P(R_N \rightarrow V|R_N P_V)(1 - P(P_V \rightarrow NV|R_N P_V)) \\
&+ P(P_V \rightarrow NV|R_N P_V)(1 - P(R_N \rightarrow V|R_N P_V))\}.
\end{aligned}
\tag{2.86}
$$

Efficiency model

$$
Q\frac{\mathrm{d}x}{\mathrm{d}t} = [S_N S_N](\infty)P(S_N \rightarrow V|S_N S_N)(1 - P(S_N \rightarrow V|S_N S_N))
$$

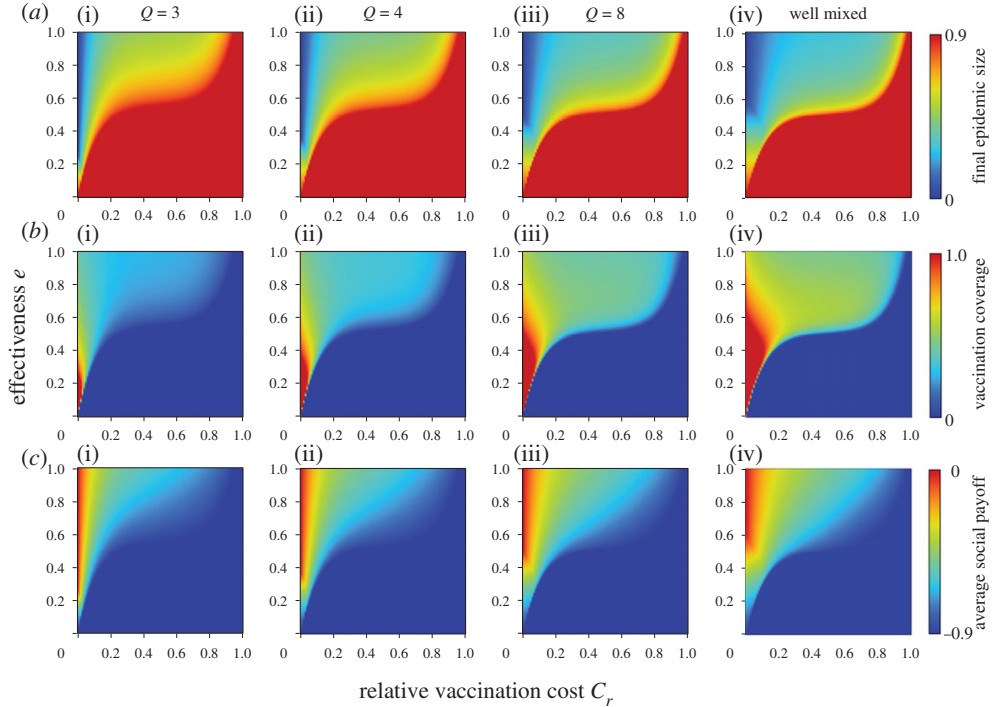

**Figure 1.** Final epidemic size (*a*), vaccination coverage (*b*) and average social payoff (*c*) for different numbers of degrees *Q*. The effectiveness model is applied. (Online version in colour.)

$$+ [S_N R_N](\infty)\{P(S_N \to V|S_N R_N)(1 - P(R_N \to V|S_N R_N))$$
$$+ P(R_N \to V|S_N R_N)(1 - P(S_N \to V|S_N R_N))\}$$
$$+ [R_N R_N](\infty)P(R_N \to V|R_N R_N)(1 - P(R_N \to V|R_N R_N))$$
$$+ [S_V S_V](\infty)P(S_V \to NV|S_V S_V)(1 - P(S_V \to NV|S_V S_V))$$
$$+ [S_V R_V](\infty)\{P(S_V \to NV|S_V R_V)(1 - P(R_V \to NV|S_V R_V))$$
$$+ P(R_V \to NV|S_V R_V)(1 - P(S_V \to NV|S_V R_V))\}$$
$$+ [R_V R_V](\infty)P(R_V \to NV|R_V R_V)(1 - P(R_V \to NV|R_V R_V))$$
$$- [S_N S_V](\infty)\{P(S_N \to V|S_N S_V)(1 - P(S_V \to NV|S_N S_V))$$
$$+ P(S_V \to NV|S_N S_V)(1 - P(S_N \to V|S_N S_V))\}$$
$$- [S_N R_V](\infty)\{P(S_N \to V|S_N R_V)(1 - P(R_V \to NV|S_N R_V))$$
$$+ P(R_V \to NV|S_N R_V)(1 - P(S_N \to V|S_N R_V))\}$$
$$- [S_V R_N](\infty)\{P(R_N \to V|S_V R_N)(1 - P(S_V \to NV|S_V R_N))$$
$$+ P(S_V \to NV|S_V R_N)(1 - P(R_N \to V|S_V R_N))\}$$
$$- [R_N R_V](\infty)\{P(R_N \to V|R_N R_V)(1 - P(R_V \to NV|R_N R_V))$$
$$+ P(R_V \to NV|R_N R_V)(1 - P(R_N \to NV|R_N S_V))\}. \tag{2.87}$$

Here, $P(A \to V \text{or} NV|AB)$ is a transition probability that the focal $A$ of pair $AB$ change to the opposite strategy (vaccinator or non-vaccinator). All transition probabilities are described in the electronic supplementary material.

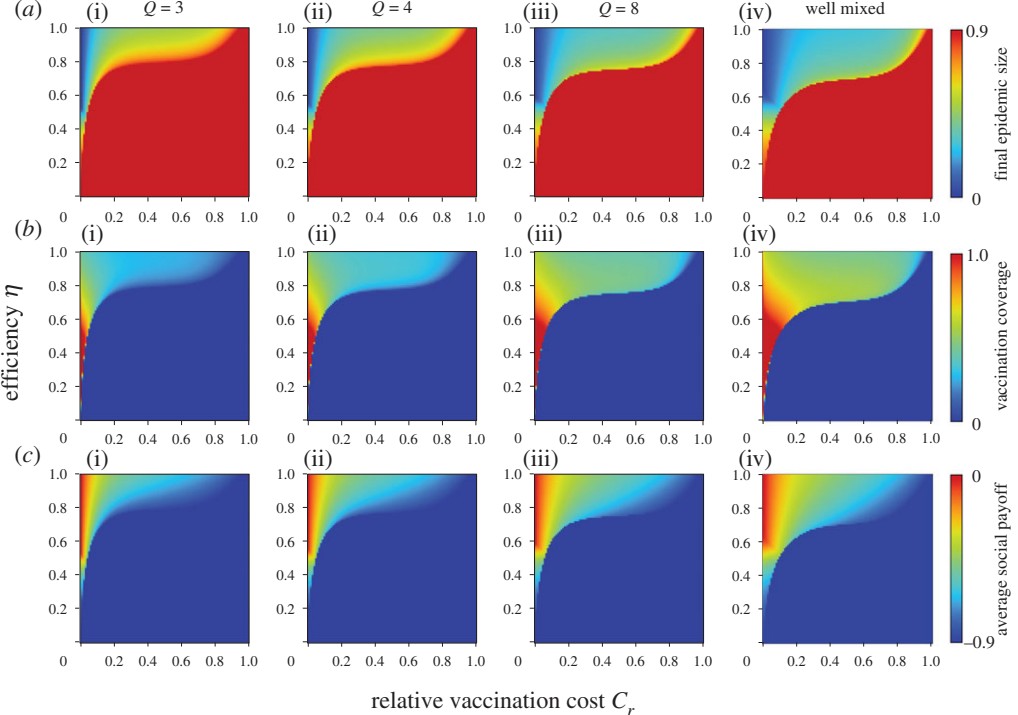

**Figure 2.** Final epidemic size (*a*), vaccination coverage (*b*) and average social payoff (*c*) for different numbers of degrees *Q*. The efficiency model is applied. (Online version in colour.)

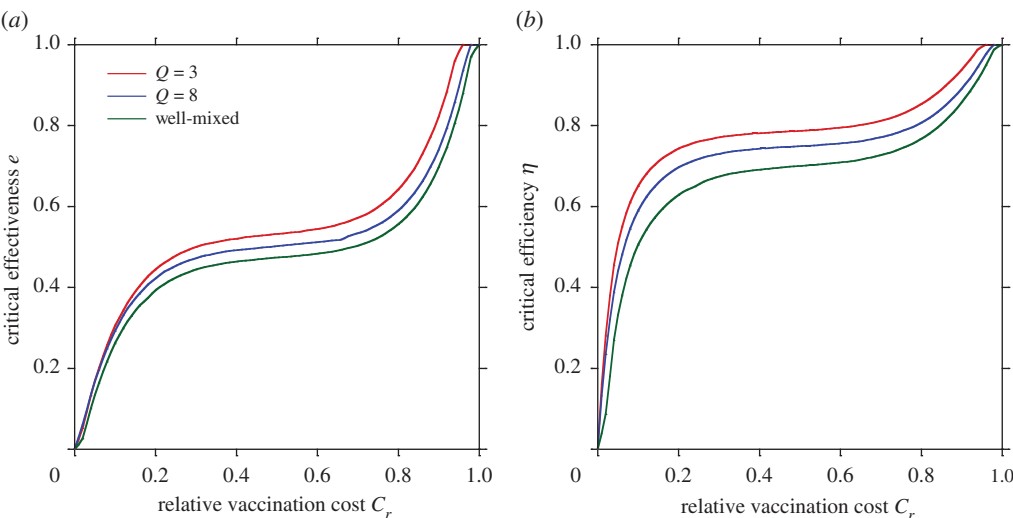

**Figure 3.** Critical effectiveness (*a*) and critical efficiency (*b*) versus relative vaccination cost $C_r$ under different numbers of degrees *Q*. (Online version in colour.)

All of the above dynamic equations can be solved numerically. Therefore, the final result is affected by a two-stage process: single-season SIR dynamics and the strategy adaptation process. We rely on an explicit scheme, and we evaluate how this specific dynamic system evolves.

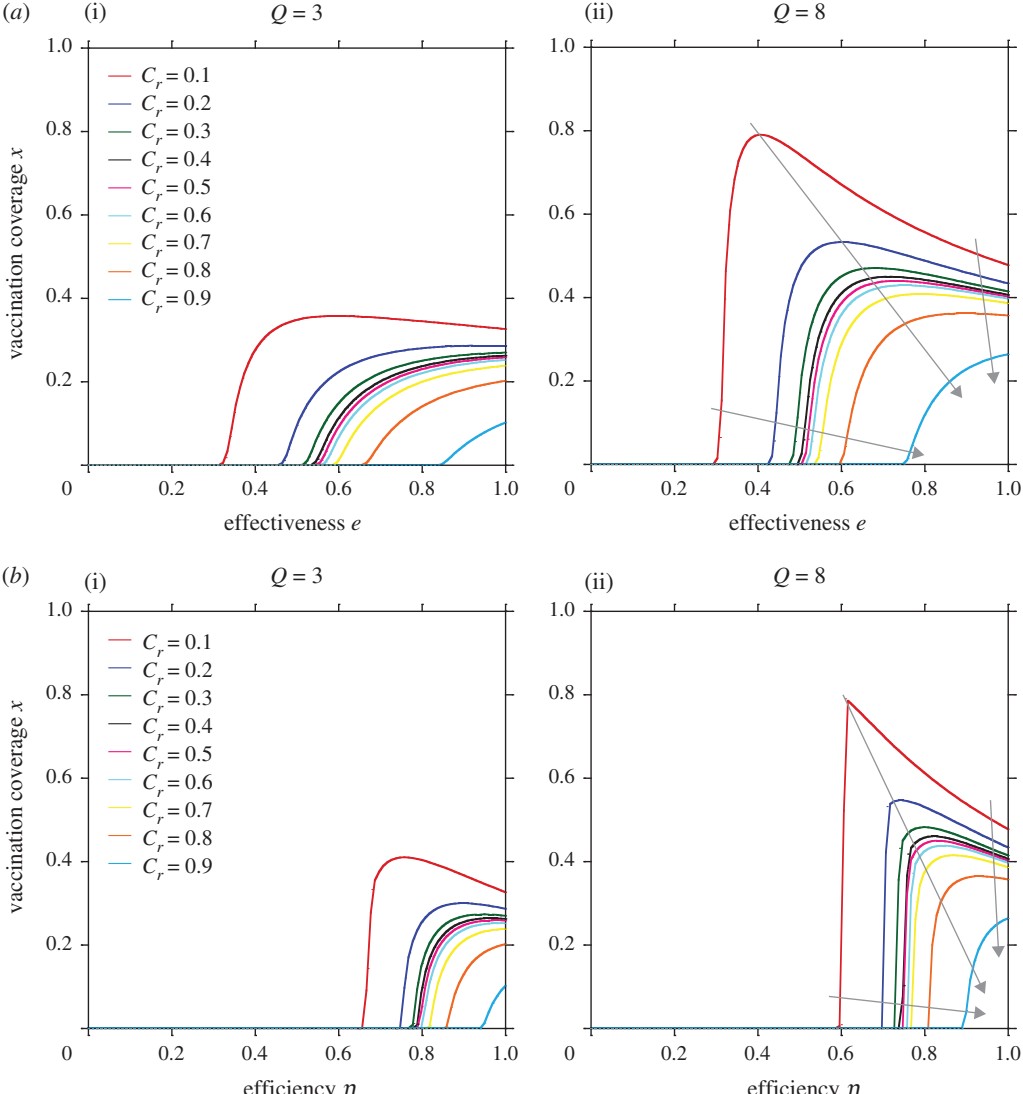

**Figure 4.** Isogram of relative vaccination cost $C_r$ drawn on a two-dimensional plane of vaccination coverage versus effectiveness (*a*) or efficiency (*b*). The grey arrows indicate the direction of the slope. (Online version in colour.)

Therefore, we can observe the final epidemic size, vaccination coverage and average social payoff in social equilibrium.

## 3. Discussion

Figures 1 and 2 show the final epidemic size (*a*), vaccination coverage (*b*) and average social payoff (*c*) corresponding to the relative vaccination cost $C_r$ and the effectiveness *e* or efficiency $\eta$, under a different number of links; degree *Q*. The panels on the far-right show previous results based on mean-field approximation (denoted 'well mixed') [20]. First, the present result with a lower degree shows a significantly different picture from the mean-field approximation prediction. A lower degree generally implies a robust environment for disease spreading and ironically leads to an individual shunning protective measures. In line with this justification, with

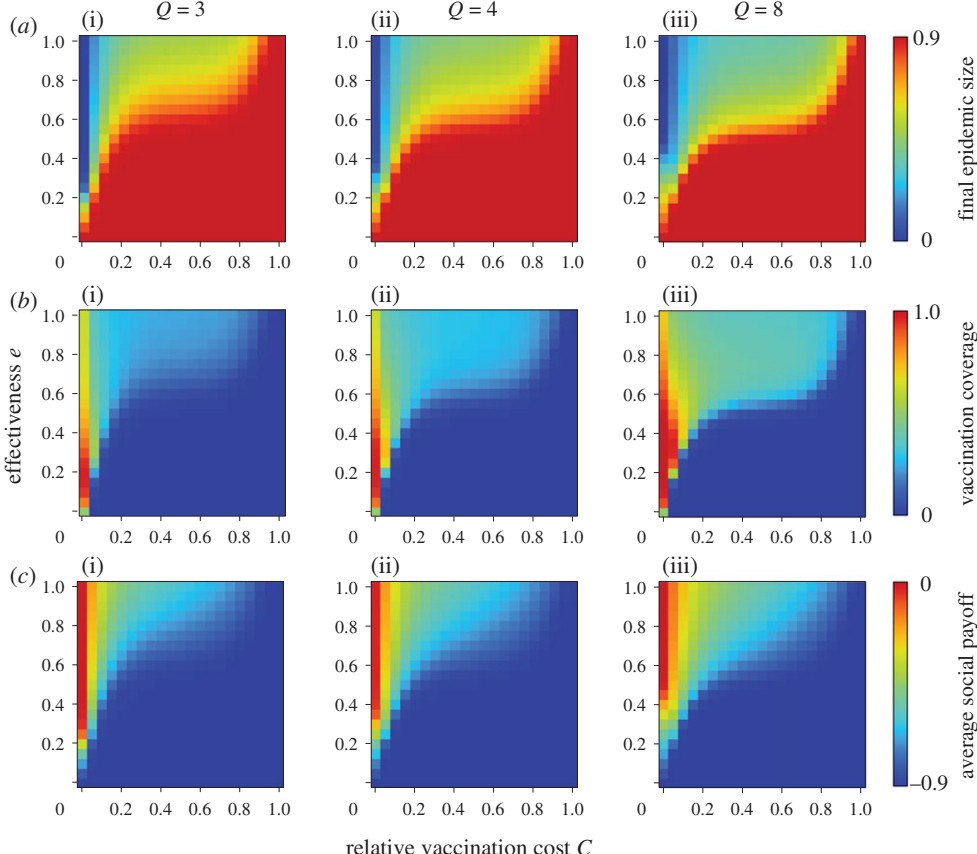

**Figure 5.** Final epidemic size (*a*), vaccination coverage (*b*) and average social payoff (*c*) for different numbers of degrees *Q*. The efficiency model is applied. (Online version in colour.)

an increase in degree, the present model's prediction gets close to the prediction of the mean-field approximation. A random regular graph with the infinite degree is quite possible because it literally means 'well mixed.' Keeping this in mind, we were able to see how meaningful the present model is over conventional analytical approaches that rely on mean-field approximation.

In all figures, the red colour in final epidemic size, blue colour in vaccination coverage and red colour in average social payoff indicate a pandemic state, in which most individuals rely on neither vaccination nor intermediate protective measures. Therefore, an almost full-scale spread of infection is inevitable. Generally speaking, these regions emerge when the reliability of vaccinations or intermediate protective measure is low or when the cost is high. The border between each of these monotone regions and the remaining area represents a threshold that suggests a combination of critical effectiveness (efficiency) and critical cost to appropriately control the epidemic spread. Therefore, under a voluntary vaccination policy, we can confirm how the relationship between the reliability of protective measures and the cost of protection is an important factor in controlling the damage caused by the epidemic spread. Figure 3 shows the relationship between cost and critical effectiveness (efficiency) under which no individual takes vaccination (or intermediate protective measures). It clearly suggests that a higher cost requires a higher effectiveness (efficiency) to enable individuals to commit because individuals have no incentive to take protective measures unless there is a sufficiently high counterbalancing effect to the high cost. An observation of the differences in the degree *Q* shows that a lower degree requires greater effectiveness (efficiency), i.e. lower level social networks are more robust against

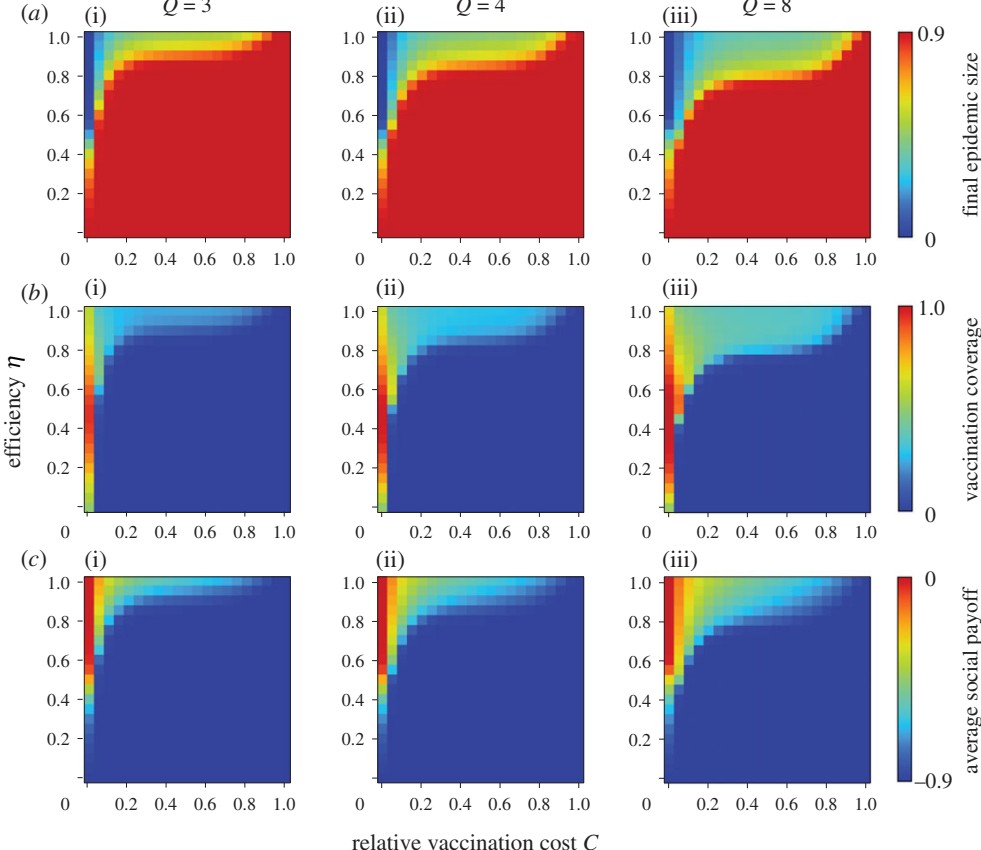

**Figure 6.** Final epidemic size (*a*), vaccination coverage (*b*) and average social payoff (*c*) for different numbers of degrees *Q*. The efficiency model is applied. (Online version in colour.)

disease spreading than those at a higher level. This ironically results in a reduced incentive for individuals to vaccinate (or take intermediate protective measures) because of the temptation to take a free ride on herd immunity.

As shown in figure 1 and 2, a lower effectiveness (or efficiency) corresponds with a higher vaccination coverage as long as the costs imposed are acceptable. This ironic situation can be explained as follows: when a protective measure is less reliable, it will create more uncertainty and fear, and more individuals will commit to the measure. However, even if a large proportion of individuals take protective measures, the epidemic cannot be eradicated if the vaccination is unreliable.

A comparison of the effectiveness and efficiency models in figures 1 and 2 show that the latter has a wider pandemic phase at first glance. This implies that intermediate protective measures with certain $\eta$ are less effective for suppressing an epidemic than imperfect vaccination, and this finding is consistent with our previous results based on mean-field approximation.

Figure 4 shows the isograms of the relative vaccination cost $C_r$ drawn on a two-dimensional plane of vaccination coverage and effectiveness (efficiency). The grey arrows indicate the direction of the rising slope of $C_r$, thus indicating that vaccination coverage (the rate of intermediate protective measures) gradually increases with decreasing effectiveness (efficiency) from $e = 1.0$ ($\eta = 1.0$) and dramatically drop down to zero at the critical effectiveness (efficiency), when the relative vaccination costs are low. This finding confirms what we have discussed above and is consistent with the reports by Wu *et al.* [18] and Chen *et al.* [21]. On the other hand, in case of high

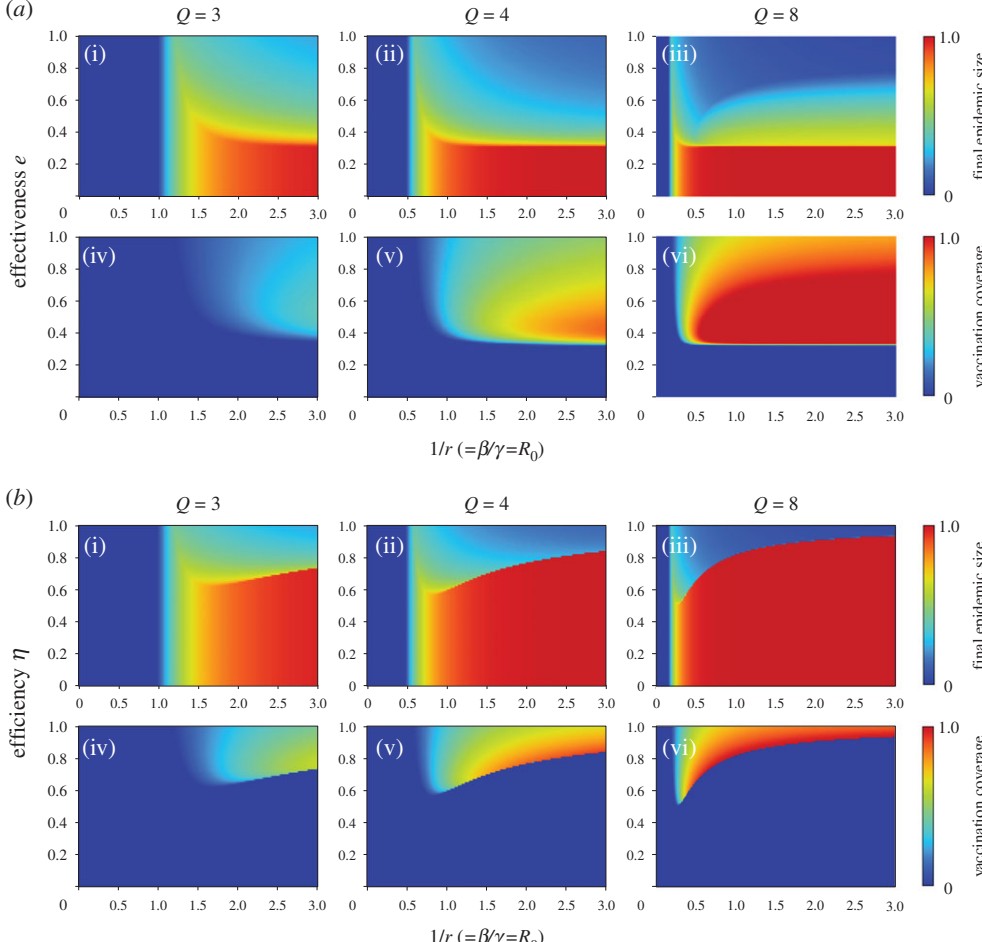

**Figure 7.** Final epidemic size and vaccination coverage as a function of the inverse of the relative recovery rate $1/r$ and the protection quality: (*a*) effectiveness $e$ and (*b*) efficiency $\eta$ for regular random networks with different numbers of degrees $Q$. We assumed $C_r = 0.1$. (Online version in colour.)

vaccination costs, the vaccination coverage gradually decreased as the effectiveness of vaccination (efficiency of intermediate protective measure) decrease from $e = 1.0$ ($\eta = 1.0$).

To validate the theoretical results, a series of numerical simulations based on the MAS approach was performed. Following the procedure in previous studies [8], we set $\beta$ as the point at which the final epidemic size exceeds the predefined threshold (i.e. 0.9) without any vaccinated individuals. Accordingly, we set $\beta = 0.72$, 0.38 and 0.14 for random regular graph with degree $Q = 3$, 4 and 8, respectively. We set $\gamma = 1/3$. A typical flu is assumed to determine these disease parameters. The results shown below were obtained by a collective average of 100 independent realizations starting from various initial conditions. The results for the effectiveness and efficiency models are shown in figures 5 and 6. Generally, all results are consistent with figures 1 and 2, although there are subtle discrepancies arising from the fact that the above simulation assumed a finite population size of $N = 10^4$.

To characterize the effect of infection rates on vaccination behaviour, final epidemic size and vaccination coverage were shown as a function of the inverse of the effective recovery rate $1/r$ and the protection quality: (i) effectiveness $e$ and (ii) efficiency $\eta$ in figure 7. We fixed the relative vaccination cost $C_r = 0.1$. Looking at the effect of the degree of $Q$ on vaccination behaviour,

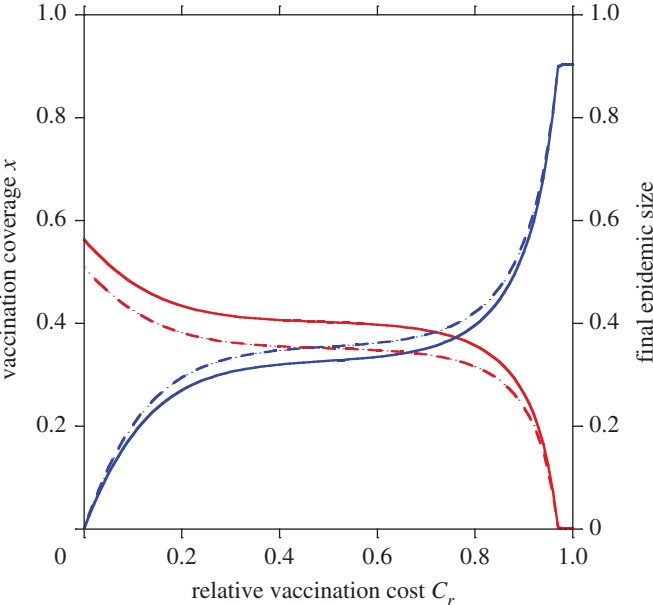

**Figure 8.** Differences of vaccination coverage and final epidemic size caused from whether to assume homogeneous or evolutionary heterogeneous distribution of vaccinator and non-vaccinator. Red lines present vaccination coverage and blue lines present final epidemic size. In the solid lines, we assumed the evolutionary heterogeneous distribution and the parameter $\alpha$ varied every strategy adaptation. On the other hand, in dash line, we assumed the homogeneous distribution ($\alpha = x(1 - x)$). We assumed perfect vaccination. (Online version in colour.)

individuals do not vaccinate until the relative recovery rate $r$ is less than the critical relative recovery rate $r_c$, as expressed in Eq. (A-19) of the electronic supplemental material, because the epidemic does not spread. Furthermore, vaccination coverage increases with an increase in $Q$, both in terms of effectiveness and efficiency. This implies that a high number of degrees promote not only the epidemic spread but also the (cooperative) vaccination behaviour. The critical value of effectiveness, below which no individual will be vaccinated, does not show sensitivity to the relative cost, which is expressed as $e = 0.35$ in this case. On the other hand, the critical value of efficiency, below which no individual will be vaccinated, increases monotonically with an increase in the relative recovery rate. This result is consistent with results by Cardillo *et al*. [14].

In general, most of theoretical vaccination game model using mean-field approximation ignored the local correlations between vaccinators and non-vaccinators and assumed a homogeneous distribution of vaccinator and non-vaccinator. On the other hand, pair approximation can capture the evolution of heterogeneous distribution of vaccinator and non-vaccinator caused by strategy adaptation. In the model formulation, we introduced new parameter; $\alpha$, meaning disassortativity at the beginning of every season, that can be time-evolved according to equations (2.86) and (2.87) when we digress from the mean-field approximation mentioned above. Let us name this setting 'heterogeneous' case. Contrariwise, when we follow to the conventional idea; taking mean-field approximation, $\alpha$ is always frozen at $\alpha = x(1 - x)$, which is called 'homogeneous' case. As shown in figure 8, when the heterogeneous setting was imposed, vaccination coverage is relatively high and final epidemic size is relatively low compared to the homogeneous one ($\alpha = x(1 - x)$). Regardless of $C_r$, we confirmed that $\alpha$ is always lower than that the homogeneous case; $x(1-x)$, because of the existence of the cluster of vaccinators and non-vaccinators. The existence of non-vaccinators cluster with high-frequency pushes up the risk of the disease spreading shared amid non-vaccinators, which makes defectors commit vaccination. This subsequently enables less final epidemic size when the heterogeneous case is presumed.

# 4. Conclusions

We presented a more precise theoretical framework of the vaccination game that considers imperfect vaccination and intermediate protective measures. The greatest advantage of our model is that it relies on a pair approximation epidemic model instead of mean-field approximation. The exact mathematical formulae for both dynamic processes, namely, epidemic spreading and strategy updating, are explicitly discussed. When solving the ODEs for the constructed pair approximation epidemic model, the critical vaccination coverage and the final fractions for each individual were derived. The results show that it is important to consider the degree effect in pairwise or mean-field approximation, particularly when a lesser degree is presumed. In addition, the effect of imperfect vaccination and intermediate protective measure strongly affects vaccination behaviour and final epidemic size. We validated our framework by comparing it with the MAS approach.

Data accessibility. All data have been generated from a series of numerical simulations. We have provided the source code (c++) for the numerical simulation in the electronic supplementary material.
Authors' contributions. K.K. developed the model, performed numerical simulations, analysed the results and drafted the manuscript. M.T. helped to design the model formulation, analysed the results and critically revised the manuscript. J.T. helped to design the study, coordinated the study and also helped draft the manuscript. All authors gave final approval for publication and agree to be held accountable for the work performed therein.
Competing interests. We declare we have no competing interests.
Funding. This study was partially supported by the Grant-in-Aid for Scientific Research (KAKENHI) from JSPS (grant no. JP 19KK0262) awarded to Professor Tanimoto.

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
