## [Peer Review File · Proceedings. Mathematical, Physical, and Engineering Sciences]

Review History

RSPA-2020-0769.R0 (Original submission)

Review form: Referee 1

Is the manuscript an original and important contribution to its field?

Excellent

Is the paper of sufficient general interest?

Excellent

Is the overall quality of the paper suitable?

Excellent

Do you have any ethical concerns with this paper?

No

Recommendation?

Accept with minor revision (please list in comments)

Comments to the Author(s)

In the paper, the authors proposed the pair approximation model for the vaccination game. And they studied the impact of imperfect vaccination and intermediate protective measures on the

epidemic spreading and obtained some interesting results.
Thus, I will recommend this paper for publication once the following comments are addressed.

Q1: The authors do not give the exact description of all the parameters in the paper. Please add the detailed description.

Q2: I don't understand that how the vaccinator-non-vaccinator connection coefficient α change because of the changing the strategy in two models. Please give the detailed description.

Q3: In line 534, I doubt that you set three different values of β for the different degrees.

Q4: The paper would benefit by considering the following research papers focusing on effects of vaccination policy on epidemic spreading. The might be useful in the instruction section and in the discussion.

[1] Effects of behavioral response and vaccination policy on epidemic spreading- an approach based on evolutionary-game dynamics, Sci. Rep. 4, 5666 (2014).

[2] Preferential imitation can invalidate targeted subsidy policies on seasonal- influenza diseases, Appl. Math. Comput. 294, 332 (2017).

Review form: Referee 2

Is the manuscript an original and important contribution to its field?

Excellent

Is the paper of sufficient general interest?

Good

Is the overall quality of the paper suitable?

Excellent

Do you think some of the material would be more appropriate as an electronic appendix?

Yes

Do you have any ethical concerns with this paper?

No

Recommendation?

Accept with minor revision (please list in comments)

Comments to the Author(s)

This manuscript provides a novel theoretical framework for the vaccination game, i.e. a social dilemma in which vaccination or prophylaxis is seen as a cooperative action that implies some cost whereas risky behavior (defection) obtains benefit from the costly actions of cooperators by a herd immunity mechanism.

As the author states in the introduction many of the approaches up to date rely on simulations or simple mean field theories that do not allow to take into account network effect, in particular when this networks are highly clustered (as it is the case of social networks). Here, the authors present a novel framework that, based on the pair approximation technique, provide an analytical framework to test the vaccination adoption in real contact networks. I find this manuscript an important contribution to the field and highly recommend its publication.

As a suggestion I recommend the authors to re-organize the manuscript and try to place some of the calculations in an appendix. Obviously all the details provided help the interested reader to replicate and apply the new formalism but, as it is, it screens some results, such as the dependence of vaccination with the mean degree of the network. I think that his problem could

be easily solved by placing some details about the formalism in an appendix. As I said, this is just a suggestion to the authors.

Another suggestion is to incorporate the recent work: Phys. Rev. Res. 2, 023181 (2020) to the discussion in the introduction, as one of the few works that provide a theoretical framework that allows to incorporate the structure of the network in this kind of disease-driven dilemmas (there through the use of Markovian equations).

Decision letter (RSPA-2020-0769.R0)

16-Dec-2020

Dear Dr Kuga,

On behalf of the Editor, I am pleased to inform you that your Manuscript RSPA-2020-0769 entitled "Pair Approximation model for the Vaccination Game: Predicting the Dynamic Process of Epidemic Spread and Individual Actions against Contagion" has been accepted for publication subject to minor revisions in Proceedings A. Please find the referees' comments below.

The reviewer(s) have recommended publication, but also suggest some minor revisions to your manuscript. Therefore, I invite you to respond to the reviewer(s)' comments and revise your manuscript. Please note that we have a strict upper limit of 28 pages for each paper. Please endeavour to incorporate any revisions while keeping the paper within journal limits. Please note that page charges are made on all papers longer than 20 pages. If you cannot pay these charges you must reduce your paper to 20 pages before submitting your revision. Your paper has been ESTIMATED to be 26 pages. We cannot proceed with typesetting your paper without your agreement to meet page charges in full should the paper exceed 20 pages when typeset. If you have any questions, please do get in touch.

It is a condition of publication that you submit the revised version of your manuscript within 7 days. If you do not think you will be able to meet this date please let me know in advance of the due date.

To revise your manuscript, log into <https://mc.manuscriptcentral.com/prsa> and enter your Author Centre, where you will find your manuscript title listed under "Manuscripts with Decisions." Under "Actions," click on "Create a Revision." Your manuscript number has been appended to denote a revision.

You will be unable to make your revisions on the originally submitted version of the manuscript. Instead, revise your manuscript and upload a new version through your Author Centre.

When submitting your revised manuscript, you will be able to respond to the comments made by the referee(s) and upload a file "Response to Referees" in Step 1: "View and Respond to Decision Letter". You can use this to document any changes you make to the original manuscript. In order to expedite the processing of the revised manuscript, please be as specific as possible in your response to the referee(s).

IMPORTANT: Your original files are available to you when you upload your revised manuscript. Please delete any redundant files before completing the submission process.

When uploading your revised files, please make sure that you include the following as we cannot proceed without these:

- 1) A text file of the manuscript (doc, txt, rtf or tex), including the references, tables (including captions) and figure captions. Please remove any tracked changes from the text before submission. PDF files are not an accepted format for the "Main Document".
- 2) A separate electronic file of each figure (tif, eps or print-quality pdf preferred). The format should be produced directly from original creation package, or original software format.
- 3) Electronic Supplementary Material (ESM): all supplementary materials accompanying an accepted article will be treated as in their final form. Note that the Royal Society will not edit or typeset supplementary material and it will be hosted as provided. Please ensure that the supplementary material includes the paper details where possible (authors, article title, journal name). Supplementary files will be published alongside the paper on the journal website and posted on the online figshare repository (<https://figshare.com>). The heading and legend provided for each supplementary file during the submission process will be used to create the figshare page, so please ensure these are accurate and informative so that your files can be found in searches. Files on figshare will be made available approximately one week before the accompanying article so that the supplementary material can be attributed a unique DOI. Alternatively you may upload a zip folder containing all source files for your manuscript as described above with a PDF as your "Main Document". This should be the full paper as it appears when compiled from the individual files supplied in the zip folder.

Article Funder

Please ensure you fill in the Article Funder question on page 2 to ensure the correct data is collected for FundRef (<http://www.crossref.org/fundref/>).

Media summary

Please ensure you include a short non-technical summary (up to 100 words) of the key findings/importance of your paper. This will be used for to promote your work and marketing purposes (e.g. press releases). The summary should be prepared using the following guidelines:

- *Write simple English: this is intended for the general public. Please explain any essential technical terms in a short and simple manner.
- *Describe (a) the study (b) its key findings and (c) its implications.
- *State why this work is newsworthy, be concise and do not overstate (true 'breakthroughs' are a rarity).
- *Ensure that you include valid contact details for the lead author (institutional address, email address, telephone number).

Cover images

We welcome submissions of images for possible use on the cover of Proceedings A. Images should be square in dimension and please ensure that you obtain all relevant copyright permissions before submitting the image to us. If you would like to submit an image for consideration please send your image to proceedingsa@royalsociety.org

Once again, thank you for submitting your manuscript to Proceedings A and I look forward to receiving your revision. If you have any questions at all, please do not hesitate to get in touch.

Best wishes
Raminder Shergill
proceedingsa@royalsociety.org
Proceedings A

on behalf of

Professor Vincenzo Capasso
Board Member
Proceedings A

Reviewer(s)' Comments to Author:

Referee: 1

Comments to the Author(s)

In the paper, the authors proposed the pair approximation model for the vaccination game. And they studied the impact of imperfect vaccination and intermediate protective measures on the epidemic spreading and obtained some interesting results.

Thus, I will recommend this paper for publication once the following comments are addressed.

Q1: The authors don not give the exact description of all the parameters in the paper. Please add the detailed description.

Q2: I don't understand that how the vaccinator-non-vaccinator connection coefficient α change because of the changing the strategy in two models. Please give the detailed description.

Q3: In line 534, I doubt that you set three different values of β for the different degrees.

Q4: The paper would benefit by considering the following research papers focusing on effects of vaccination policy on epidemic spreading. The might be useful in the instruction section and in the discussion.

[1] Effects of behavioral response and vaccination policy on epidemic spreading- an approach based on evolutionary-game dynamics, *Sci. Rep.* 4, 5666 (2014).

[2] Preferential imitation can invalidate targeted subsidy policies on seasonal- influenza diseases, *Appl. Math. Comput.* 294, 332 (2017).

Referee: 2

Comments to the Author(s)

This manuscript provides a novel theoretical framework for the vaccination game, i.e. a social dilemma in which vaccination or prophylaxis is seen as a cooperative action that implies some cost whereas risky behavior (defection) obtains benefit from the costly actions of cooperators by a herd immunity mechanism.

As the author states in the introduction many of the approaches up to date rely on simulations or simple mean field theories that do not allow to take into account network effect, in particular when this networks are highly clustered (as it is the case of social networks). Here, the authors present a novel framework that, based on the pair approximation technique, provide an analytical framework to test the vaccination adoption in real contact networks. I find this manuscript an important contribution to the field and highly recommend its publication.

As a suggestion I recommend the authors to re-organize the manuscript and try to place some of the calculations in an appendix. Obviously all the details provided help the interested reader to replicate and apply the new formalism but, as it is, it screens some results, such as the dependence of vaccination with the mean degree of the network. I think that his problem could be easily solved by placing some details about the formalism in an appendix. As I said, this is just a suggestion to the authors.

Another suggestion is to incorporate the recent work: *Phys. Rev. Res.* 2, 023181 (2020) to the discussion in the introduction, as one of the few works that provide a theoretical framework that allows to incorporate the structure of the network in this kind of disease-driven dilemmas (there through the use of Markovian equations).

Board member pre-assessment comments (if available):

Board Member: 1

Comments to Author(s):

The paper is acceptable conditional upon an accurate revision as from Reviewers' reports.

The authors are invited to include a file "Responses to Reviewers", in which they answer to each question raised by the Reviewers in the style

Q_n:....

A_n...

.

Decision letter (RSPA-2020-0769.R1)

12-Jan-2021

Dear Dr Kuga

I am pleased to inform you that your manuscript entitled "Pair Approximation model for the Vaccination Game: Predicting the Dynamic Process of Epidemic Spread and Individual Actions against Contagion" has been accepted in its final form for publication in Proceedings A.

Our Production Office will be in contact with you in due course. You can expect to receive a proof of your article soon. Please contact the office to let us know if you are likely to be away from e-mail in the near future. If you do not notify us and comments are not received within 5 days of sending the proof, we may publish the paper as it stands.

COVID-19 rapid publication process: We are taking steps to expedite the publication of research relevant to the pandemic. If you wish, you can opt to have your paper published as soon as it is ready, rather than waiting for it to be published on the scheduled Wednesday.

This means your paper will not be included in the weekly media round-up which the Society sends to journalists ahead of publication. However, it will appear in the COVID-19 Publishing Collection which journalists will be directed to each week

(<https://royalsocietypublishing.org/topic/special-collections/novel-coronavirus-outbreak>)

If you wish to have your paper published immediately please notify proca_proofs@royalsociety.org and press@royalsociety.org

The Royal Society has signed a Wellcome Trust statement on the subject of research findings and data relevant to the coronavirus (COVID-19) outbreak. We are one of several signatories to this statement and our collective aim is to ensure that the relevant research and data are shared rapidly and openly in order to inform the worldwide public health response and to help save lives. We are therefore making papers related to COVID-19 open access free of charge.

Under the terms of our licence to publish you may post the author generated postprint (ie. your accepted version not the final typeset version) of your manuscript at any time and this can be made freely available. Postprints can be deposited on a personal or institutional website, or a recognised server/repository. Please note however, that the reporting of postprints is subject to a media embargo, and that the status the manuscript should be made clear. Upon publication of the definitive version on the publisher's site, full details and a link should be added.

You can cite the article in advance of publication using its DOI. The DOI will take the form: 10.1098/rspa.XXXX.YYYY, where XXXX and YYYY are the last 8 digits of your manuscript number (eg. if your manuscript number is RSPA-2017-1234 the DOI would be 10.1098/rspa.2017.1234).

For tips on promoting your accepted paper see our blog post:
<https://royalsociety.org/blog/2020/07/promoting-your-latest-paper-and-tracking-your-results/>

On behalf of the Editor of Proceedings A, we look forward to your continued contributions to the Journal.

Sincerely,
Raminder Shergill
proceedingsa@royalsociety.org